# Scholarship Suppression: Theoretical Perspectives and Emerging Trends

**Sean T. Stevens [1], Lee Jussim [2,*] and Nathan Honeycutt [2]**

[1]  The Foundation for Individual Rights in Education, Philadelphia, PA 19106, USA; sean.stevens@thefire.org
[2]  Department of Psychology Rutgers, The State University of New Jersey—New Brunswick, New Brunswick, NJ 08901-8554, Canada; nathan.honeycutt@rutgers.edu
[*]  Correspondence: jussim@psych.rutgers.edu

**Abstract:** This paper explores the suppression of ideas within an academic scholarship by academics, either by self-suppression or because of the efforts of other academics. Legal, moral, and social issues distinguishing freedom of speech, freedom of inquiry, and academic freedom are reviewed. How these freedoms and protections can come into tension is then explored by an analysis of denunciation mobs that exercise their legal free speech rights to call for punishing scholars who express ideas they disapprove of and condemn. When successful, these efforts, which constitute legally protected speech, will suppress certain ideas. Real-world examples over the past five years of academics that have been sanctioned or terminated for scholarship targeted by a denunciation mob are then explored.

**Keywords:** free speech; academic freedom; free inquiry; censorship; conformity; moral panics; witch hunts; heresy

## 1. Introduction

> "Protection, therefore, against the tyranny of the magistrate is not enough; there needs protection also against the tyranny of the prevailing opinion and feeling; against the tendency of society to impose, by other means than civil penalties, its own ideas and practices as rules of conduct on those who dissent from them . . . "

> -John Stuart Mill (1859)

The suppression of a scholarship is well documented throughout human history. Scholars have faced censure, or worse, because they were unpopular with political and/or religious authorities, challenged majority public opinion, or researched a taboo idea. For example, in 399 B.C. Socrates was found guilty of "corrupting the youth of Athens" and sentenced to death. Galileo was declared a heretic by the Catholic Church in the early 17th century for his support of heliocentrism and ordered to abstain completely from teaching, defending, or even discussing it. He was later convicted of heresy a second time and placed under house arrest for the rest of his life. Other forms of scholarship suppression are more subtle and are driven by fear of the social consequences one may face for pursuing controversial scholarship. Nicolaus Copernicus and Charles Darwin both delayed the publication of their most influential work (Copernicus, until after his death; Darwin for decades), presumably to avoid sanction and punishment.

Socrates is now revered by many across the world, and heliocentrism and Darwinian evolution are widely accepted. Scholarship suppression today rarely takes the form of legal government censorship, at least in Western democracies. Although one might justifiably claim that there are very few topics that can currently be considered silenced within academia, there are indeed some. Some scholars have been so severely punished and stigmatized for reaching conclusions regarding certain topics that we

will not even mention those topics in this article because we fear the social sanctions that we might be subjected to, such as inciting mobs that could get us fired. This is not a rhetorical flourish, it is a real, bona fide risk because such mobs have successfully gotten other academics fired (see [1], for a list of academics sanctioned by other academics for expressing ideas).

We start, therefore, by putting aside those taboo topics. In this paper, we first discuss freedom of expression and its relationship to academic freedom (or free inquiry). This discussion explores the differences between the legal protections provided for freedom of expression, and the moral reasons for maintaining robust protections for academic freedom. Next, we discuss the mechanisms that help suppress controversial scholarship, the social and psychological factors that motivate and sustain their deployment, and how over time they can inculcate a culture of self-censorship. Finally, we review emerging trends in scholarship suppression.

## 2. Sources of Scholarship Suppression in Academia

### 2.1. Three Forms of Suppression

Scholarship suppression occurs in three different ways. External scholarship suppression occurs when the attempt comes from forces outside the academy (interest groups, the government, religious groups, etc.). Internal scholarship suppression occurs when the attempt comes from elements within the academy, either from students, other scholars, or from college administrators. Finally, a hybrid form of scholarship suppression occurs when external sources of scholarship suppression motivate internal forces within the academy to actively suppress scholarship.

### 2.2. Internal Suppression Is the Most Severe Form

Although we will touch on all three forms of scholarship suppression, this paper is primarily focused on how internal academic forces can foster the development of a culture of censorship among scholars, hindering the investigation of certain topics that are widely considered controversial or taboo. We consider internal suppression the most dangerous, most toxic form of research suppression in the U.S. and democratic west for two reasons (which is the focus of the present paper; government censorship in authoritarian regimes would require an entirely separate paper). First, in the case of hybrid instances, although outsiders might instigate a suppression process, the power (e.g., to fire, punish, and retract an article) always rests with insiders; the key actors and decision-makers are the insiders, not the outsiders.

Second, and even more important, there are reasons to believe that internal scholarship suppression is far more powerful than external or hybrid scholarship suppression. The key idea here is that academic success hinges on the views of other academics, whereas external efforts to suppress are unusual events and can thus be considered outliers. Although there are cases in which rightwing media have instigated outrage mobs that have led to sanctions of faculty [2], such instances are usually for things like incendiary tweets (in the referenced case here, "All I want for Christmas is white genocide"), rather than the conventional expression of academic ideas in books, colloquia, and peer reviewed publications. In the course of normal academic events, faculty rewards and punishments for scholarship in the U.S., for example, literally hinge to no extent whatsoever on the opinions of Fox News viewers.

Although academia should continue to remain on-guard for external sources of scholarship suppression, internal sources of scholarship suppression are very real, more effective, and deserve more attention so that they can be better guarded against. We note here that all of the authors are American citizens and we are most familiar with the issues of free expression and academic freedom in the United States. So, except where otherwise noted, our subsequent comments are largely restricted to issues of free expression and academic freedom within the United States.

## 3. Freedom of Expression and Academic Freedom

We define free expression as the refusal to allow individual expression or thought to be controlled, without consent, by an external authority (see also [3]). We consider the concept of freedom of expression as a superordinate category that includes the domains of religion, speech, association, and inquiry. Each of these freedoms protects the individual from punishment for speech or being compelled to express themselves in a particular way. For example, freedom of religion allows individuals to freely choose the religion they practice and how often they observe religious customs. Individuals can also choose not to practice any religion without fear of punishment.

### 3.1. Freedom of Expression and the Law

In Western democracies today it is fairly easy to take freedom of expression for granted. However, it took centuries for a right to freedom of expression to be widely recognized and written into law. The first formal legal protections emerged with the passage of the Bill of Rights in England in 1689, the adoption of the Declaration of the Rights of Man and of the Citizen during the early days of the French Revolution in 1789, and the passage of the First Amendment to the United States Constitution in 1791. The latter currently provides robust legal protection for the freedom of expression, restricting the U.S. government's ability to regulate speech, religious practice, the press, people's ability to assemble, and people's ability to petition the government for a redress of grievances. Court rulings have further clarified that the speech protected by the First Amendment also encompasses non-verbal forms of expression such as artwork [4] or even behavior such as defacing an American flag [5] or burning one [6].

Although the First Amendment protects citizens from restrictions on expression imposed by the government, it does not offer protection from private actors (e.g., other individuals, a community, a corporation, or private organization). There are at least two implications of this. First, it means that public colleges and universities are required to abide by the First Amendment. Private colleges and universities are not, yet many have policies or bylaws that effectively require them to do so [7]. Second, other private actors are protected from the government restricting their ability to express opposition to another's protected speech, provided they do this peacefully.

For instance, the First Amendment protects a group of Black Lives Matter activists on the campus of a state university who silently protest an academic panel on the limits and downsides of identity politics by holding up signs during the discussion or turning their backs on the speakers. However, if any members of the group loudly shouted down the speakers so that they could not continue their discussion, the university would be expected to intervene so the discussion could continue. This latter form of protest behavior is referred to as the heckler's veto, and courts in the U.S. consistently rule that such behavior is a violation of the First Amendment rights of the shouted down speaker or speakers [8].

### 3.2. Freedom of Inquiry and Belief, and Their Limitations

Legal protections of speech, whether in the U.S., U.K., or elsewhere, generally do not explicitly mention freedom of inquiry. However, since the mid-20th century university faculty in many countries have possessed some degree of academic freedom, a protection that grants scholars the freedom to research, teach, or communicate facts or ideas without fear of suppression or censure, job loss, or imprisonment. A formal definition of academic freedom was adopted by the American Association of University Professors (AAUP) in the 1940 Statement of Principles on Academic Freedom [9]. This statement contends that academic freedom is essential to the common good and the search for truth, and delineates what represents acceptable academic practice in the public square, in one's research, and in the classroom.

Even though American courts have recognized a relationship between academic freedom and the First Amendment, the two are distinct legal concepts [10]. The First Amendment applies to all citizens and affords them protection from government restrictions on their expression. The concept of

academic freedom is more narrowly focused on protecting the ability of faculty and students within academia to engage in the free and open inquiry of their individual scholarly interests and pursuits, with little to no restriction. Faculty are also granted considerable latitude in how they teach, provided they demonstrate professional competence and avoid introducing controversial material that is not related to the course.

A few examples can help us demonstrate when academic freedom protects a scholar and when it does not. First, consider a hypothetical case of an individual who sends out a series of tweets extolling the flat Earth theory and urges people to join them as a member of the Flat Earth Society. This individual also happens to be a professor of French literature at a state university; however there is no mention of this affiliation or even of being a professor in the individual's profile. The series of tweets goes viral and someone identifies the individual as a professor of French literature. A campaign is organized with the goal of getting the professor fired from the university for their belief in the flat Earth theory and membership in the Flat Earth Society. The professor, however, has never discussed the flat Earth theory in the classroom. In this situation the university can treat this professor as someone who holds a very curious belief and who associates with some very curious people. However, it would probably be difficult for the college or university to sanction the professor for beliefs they expressed as a private citizen, because the professor never claimed they were speaking on behalf of the college or university.

However, if this professor decided to devote classroom time in a French literature course to a discussion of the flat Earth theory, the university could implement sanctions. Advocating the flat Earth theory is entirely irrelevant to teaching French literature. Therefore the professor could be sanctioned for failing to meet standards of professional competence.

Now, consider if a discussion of Flat Earth theory occurred in the professor's French literature course because it was mentioned in an assigned reading and a student who is unfamiliar with the theory has asked what it contends and if there is any basis for it. In this scenario, because the discussion was prompted by a student question, it is unlikely that the university could sanction the professor for a lack of professional competence. Even though the professor is someone who holds a very curious belief and associates with some very curious people, their decision to answer a student's question about Flat Earth Theory would likely be protected by their academic freedom.

Finally, consider a professor of geology at the same state university. Just like the French literature professor, this geology professor believes in the flat Earth theory and is a member of the Flat Earth Society, and a campaign is organized with the goal of getting the professor fired from the university. However, compared to the French literature professor this is a much different situation. Unless the geologist can provide evidence demonstrating the possibility that the Earth is flat that is also accepted by some other credible geologists (credibility being established, for example, by peer reviewed publications), the university may be able to sanction them for failing to meet standards of professional competence. This is because flat Earth theory has been resoundingly rejected by the professor's field of (purported) expertise. If this professor went even further and actually devoted classroom time to lecturing students about the veracity of the flat Earth theory then academic freedom would not offer them any protection and the university may be able to fire them for failing to meet standards of professional competence.

Thus, faculty are permitted to hold all sorts of strange beliefs without sanction as long as: 1. they are irrelevant to their professional expertise and not brought into the classroom gratuitously or 2. they can actually justify strange or unorthodox beliefs in their area of expertise by conventional standards used in the faculty member's field of expertise. On the other hand, if they bring claims that are both irrelevant and unorthodox into their teaching, they are subject to sanctioning; and if they make completely unorthodox beliefs that cannot be justified by conventional methods in their field, they can be sanctioned.

### 3.3. Much of Modern Scholarship Suppression in the Academy Does not Involve Legal Issues

The perspective articulated here argues that threats to free inquiry in the academy rarely involve free speech as a legal issue. This issue is important, because we suspect that many people think of "free speech" in narrow, legalistic terms. If the government is not restricting speech, then people sometimes seem to believe there is no "free speech" issue at stake. There may be no legal issue at stake, but speech can still be threatened, ideas suppressed, and inquiry restricted. We argue herein that there are often free speech and academic freedom issues at stake, even in the absence of legal issues. Historically, for example, during the McCarthy era, many academics were fired by private universities after being questioned by the government panels inquiring into their associations with communists [11]. This did not violate the U.S. constitutional protections against government prohibitions against speech or association. Nonetheless, it clearly functioned to suppress certain ideas. Thus, a major theme throughout this paper is that threats to free speech and academic freedom can and do occur entirely legally. They occur through social, informal, organizational, and normative processes; as such, these processes can operate entirely without infringing on protections against government interference in speech.

### 3.4. Freedom of Speech as a Moral Principle

Some have argued that there really is no such thing as free speech or freedom of expression (see [12]), a contention that is more convincing than it seems at first. Freedom of expression is a value, and contentious issues about it are far more likely to arise in communities or societies where it is highly valued. Yet all societies, even the ones that consider freedom of expression one of their most important values, place some legal limits on expression. Thus, debates about free expression are about identifying the boundaries of acceptable free expression in a community or society (e.g., [13–15]), not whether a given person or group can literally express themselves. To be clear, by expression we mean the verbal (e.g., speech), written (e.g., opinion piece), artistic (e.g., political cartoon), or behavioral (e.g., flag burning) expression of a thought or viewpoint.

In *On Liberty*, Mill [16] asks to what extent society can exercise power over an individual, and he makes a very strong defense of freedom of expression with few limits. Mill contends that the expression of any idea, no matter how immoral, should be allowed unless it immediately and directly harms someone. Mill does not explicitly define harm. However, he does attempt to delineate where the boundary of acceptable expression lies by contrasting a written opinion accusing corn dealers of starving the poor with someone stating such an opinion directly to an angry mob right outside a corn dealer's house. According to Mill the written opinion, although offensive, should not be restricted, but the incitement of an angry mob that could immediately become violent with a corn dealer should be.

This distinction—between an offensive opinion that is unlikely to immediately cause direct harm to someone and an offensive opinion that could immediately incite physical violence—is important because it demonstrates that Mill did not consider offensiveness on its own as harmful. This view may be rooted in Mill's concerns about the subjectivity of people's opinions and the tendency for many to imbue those opinions with a sense of infallibility ([16] p. 17, emphasis in the original):

> *"We can never be sure that the opinion we are endeavoring to stifle is a false opinion; and if we were sure, stifling it would be an evil still. First: the opinion which it is attempted to suppress by authority may possibly be true. Those who desire to suppress it, of course deny its truth; but they are not infallible. They have no authority to decide the question for all mankind, and exclude every other person from the means of judging. To refuse hearing an opinion, because they are sure it is false, is to assume that their certainty is the same thing as absolute certainty. All silencing of discussion is an assumption of infallibility".*

In other words, Mill does not consider offensiveness an adequate basis for restricting expression, and we should not treat any single person's opinion or viewpoint as certain or infallible.

Feinberg [17] disagreed with Mill and argued that the offensiveness of an expressive action should be considered when considering restrictions on free expression. This expands the pool of expressive actions that can be restricted to those that can cause an unpleasant psychological state such as anger or disgust. Indeed, human communities and societies regulate all kinds of non-harmful, but offensive behavior even though they restrict individual liberty. For instance, most societies have legal prohibitions against two consenting adults engaging in sexual intercourse in public. This expressive action does not cause any harm, in the way Mill would define it, but it is an action that clearly possesses the potential to cause a strong negative reaction among people who happen to witness the event, and the couple would likely face legal and social sanctions as a result of their action.

Although we recognize that human communities and societies restrict all kinds of non-harmful, offensive behavior because they transgress certain moral sensibilities, we return to Mill's argument that no person's opinion or viewpoint should be treated as infallible. Indeed, human history is replete with examples of despots, dictators, and dogmatic regimes that suppressed dissent and persecuted "heretics" for their beliefs [18]. Investigations of sociopolitical tolerance are also not encouraging—people often espouse strong support for general tolerance and freedom of expression in the abstract, but this support drops, sometimes precipitously, when people are asked about individuals or viewpoints that they strongly oppose (see e.g., [19–24]). For example, Erskine [19] reviewed public opinion data on attitudes toward freedom of speech collected between 1936 and 1970. Table 1 below is reproduced from that analysis and demonstrated that a notable portion of American citizens endorsed non-specific limitations on freedom of speech and that the majority opposed freedom of speech for extremists.

**Table 1.** Support for freedom of speech 1938–1970, from Erskine (1970).

| Maximum Percentage Believing in | Before 1950 | 1950–1960 | After 1960 |
| --- | --- | --- | --- |
| Theoretical freedom of speech | 97% | Not asked | Not asked |
| Freedom of speech with non-specific limitations | 68% | 70% | 61% |
| Freedom of speech for extremists | 49% | 29% | 21% |

The dramatic drop-off from theoretical support for freedom of speech (a statement such as, "The minority should be free to criticize majority decisions"—[22]) to concrete support (e.g., for "extremist" groups as shown) was found consistently over several decades. When people are asked if they would tolerate certain kinds of expression, who or what is being granted tolerance matters a great deal (see [20,21,23,24]). By whom we mean the individual or group of individuals responsible for an expressive action. By what we mean the specific thought or viewpoint itself (e.g., support for communism), independent of who is expressing it.

Importantly, people are likely to oppose freedom of expression if they dislike the target, consider the expressive content a normative violation, and believe that the target's group is growing in strength and influence, constituting an existing threat (see [25]). For instance, support for allowing communists to freely express their ideas among American citizens has fluctuated quite a bit over time (see [15,18,19,23,24]). In 1954, only 27% of Americans said they would allow an admitted communist to make a speech in their community and an even smaller percentage (5%) said he should be allowed to keep his job teaching at the local college [26]. In 2018, 67% of Americans said a communist should be allowed to speak in their community and 61% said he should be allowed to keep his job at the local college [27]. This change in attitudes toward communists can be explained by recognizing that 1954 was the height of McCarthyism and the second Red Scare. With the emergence of the Soviet Union as a rival global superpower to the United States, communism was considered a global threat to democracy that was also growing in strength and influence [26]. With the fall of the Soviet Union in the 1990s the threat of communism has waned, and today the majority of Americans do not appear to consider communism a threat.

History, law, and philosophy converge on raising the following rhetorical question: who is so infallible that they get to decide who or what requires restriction, outright censorship, or worse?

We consider the principle that few claims are so settled as to justify censorship one of the foundational pillars of science. If some claim is definitively true (e.g., the Earth is round), then claims that the Earth is flat can be readily admitted into discourse; they will simply be easily debunked. In the social sciences, however, few claims are as certain as "the Earth is round". Our capability to fully understand the world is limited and our understanding of the world is constantly evolving. The history of science is littered with beliefs once held as all but "certain" that were subsequently rejected, such as a young Earth, a geocentric universe, leeches helped release "bad blood" or stress causes ulcers. Therefore, we contend that scholars need to be able to engage in unfettered freedom of inquiry.

### 3.5. Suppression Versus Rejection

In scholarship and science there is a difference between suppression and rejection. Suppression occurs when the fear of social sanctions prevents ideas from being explored or empirical findings from being presented in scientific or public forums. In science, rejection occurs when an idea has been explored and the evidence has been found wanting. The history of science is replete with rejected ideas, such as a geocentric solar system, young Earth, spontaneous generation of life, and the phlogiston theory of air. These ideas were thoroughly explored and rejected because the evidence available overwhelmingly disconfirmed them.

In contrast, suppression prevents an idea even from being explored. Historically, this has occurred for a wide variety of reasons, including that the idea constitutes religious heresy [18], political anathema [28], or premature canonization of the wrong idea [29]. Premature canonization refers to widespread scientific belief in a false conclusion, which leads to suppression masquerading as rejection. A classic relatively recent case of premature canonization involves the scientific identification of causes of ulcers. In the 1950s and 1960s the medical establishment had converged on the conclusion that stress caused ulcers, and a huge, lucrative pharmaceutical industry was built around treating ulcers by treating stress. When Barry Marshall came along in the 1980s producing study after study showing that bacteria, not stress, caused ulcers, he was generally dismissed as a crank and had difficulty getting the work published or treated seriously at medical conferences (the history is told in [30]). This is suppression, not "rejection" because no one ever actually refuted his research. Although the medical community was eventually persuaded by Marshall's work (indeed, he received a Nobel Prize for it), it took decades because his early work was effectively suppressed. Loeb [31] presents several examples in which premature canonization of erroneous claims unnecessarily delayed progress in astronomy. These are generally similar to the ulcers case in that ideas based on little or no evidence somehow became widely accepted, leading to initial suppression rather than refutation on ostensible grounds that what turned out to be the truth (which ran against consensus) was not credible.

To recap, rejection means science has extensively examined some claim or hypothesis and determined it to be false. In contrast, suppression means an idea either cannot be explored, or, if explored and empirically supported, is blocked from communication with the scientific community or public. In principle, unfettered free inquiry includes the possibility of reviving long-rejected ideas. However, scientists who decide to attempt to revive a long-rejected view in some field should be aware of at least two things: 1. they are going to need extraordinary evidence to persuade other scientists and 2. they are likely to receive harsh and intense criticism, and, at least initially, have their work suppressed in the sense described in this paper. This is because, even if they are right, others will likely assume they are quacks and dismiss the claims out of hand, without scrutiny.

## 4. Academic Outrage Mobs: A Theoretical Perspective on Scholarship Suppression

### 4.1. Academia Is a Social Reputational System

The production of ideas and knowledge in academia generally hinges almost entirely on the subjective evaluations of one's academic colleagues. After briefly justifying the idea that academia is a

social reputational system, we focused on how and why this renders academics particularly vulnerable to idea suppression.

All or nearly all academic incentives are fundamentally social, rather than objective:

- Admissions to graduate school? Letters of recommendation are required and important.
- Peer review? The evaluation of your work by peers.
- Grants? Usually obtained by peer review.
- First job? Peer reviewed publications and letters of recommendation, preferably from famous faculty.
- Tenure? Peer reviewed publications, grants, and letters of support from prominent faculty.
- Further promotions? Peer reviewed publications, grants, and letters of support from prominent faculty.

Since social evaluations are so central to success in academia, it is easy to induce fear of social sanctions for expressing the ideas that, though not necessarily shown to be factually or scientifically wrong, are widely unpopular or disapproved.

Suppression can occur in a variety of ways and for a wide variety of reasons, not all political. Eminent and prestigious scholars are often gatekeepers (editors, society officers, etc.) and can have outsized influence on which ideas are cultivated, ignored, or outright blocked. For example, one of the sources of psychology's Replication Crisis is that failures to replicate famous scientists' work often ran into difficulty getting published [32]. This occurred because replication attempts often targeted famous and influential articles and findings for good reasons. There may be few more important replications than those verifying the validity of the claims made based on original studies widely viewed as groundbreaking. Typically, however, the replication attempt would be sent to the original authors for review, because the original authors would be viewed as highly expert in the research area and specific methods having already published on that topic using those same methods.

However true that may be, this process also creates a built-in conflict of interest: If a scholar's success and prestige hinged in part on the accolades and respect from colleagues that accrued as a result of the original paper and findings, and if they are sent a paper that failed to replicate their original findings, they have ample incentives to suppress the failed replication, e.g., by producing a scathing review to the editor rejecting the paper. Of course the criticisms will be framed in purely "scientific" terms—methodology, statistics, logical inference, etc.—rather than personal terms. Regardless, few editors indeed are likely to be willing to accept a paper over the adamant rejections and intense criticisms of famous, eminent experts. Doing so might put their own careers at risk.

This dynamic probably helps explain why some findings and claims in psychology have more appearance than reality of scientific credibility. The appearance is created by the "scientific" literature, which, because of suppression of failed replications, is populated disproportionately by papers confirming the effect. In short, if the famous and eminent block publication of findings contesting that upon which they built their fame and eminence, it may never see the light of day, at least not until they are long gone.

Of course, this dynamic is likely far broader than merely failed replications. Any time research produces either findings or conclusions that some large interest group in that field opposes, a similar dynamic can function to suppress those findings. The same dynamic can play out with respect to grants, which greatly facilitate the conduct of research on various topics (and in some cases are an absolute necessity). However, they are also highly competitive, with government funding rates in the social sciences in the U.S. often around 20% [33]. This means that one needs to usually receive favorable ratings from almost every reviewer to get funded. If the ability to do so is rendered more difficult because even a substantial minority of one's colleagues object to the type of work being proposed, that work is unlikely to get funded or conducted. The work is effectively suppressed. Due to the social reputational nature of academia, the ideas of academics are especially vulnerable to suppression through social ostracism and punishments.

### 4.2. Freedom of Expression Versus Freedom of Inquiry

Although many people seem to assume that freedom of inquiry and academic freedom are subsets of, and largely subsumed by, the legal protections for freedom of speech, the two are distinct legal concepts [34]. Furthermore, free expression, whether protected on the grounds of the First Amendment or supported because of belief in free expression as a moral principle (see above), can infringe on someone else's academic freedom and their ability to engage in free inquiry. The prototypical case is denunciation: In the U.S., the government (because of free speech guarantees) cannot prevent someone from denouncing others as evil and calling for them to be sanctioned. Such denunciations can suppress not only the speech of those denounced, but others who agree with the denounced. Thus, free inquiry is not a subset of freedom of expression, and one cannot assume that, if free speech is protected, ipso facto, free inquiry is protected. The two can and do come into conflict.

The basic recipe for this conflict in academic, or other contexts, is simple. First, someone schedules an event (e.g., a conference; an invited talk), publishes an article (peer-reviewed or op-ed), or states something that another person, or group of people, considers offensive or harmful. Then the aggrieved person or group of people organizes in some way (e.g., by petitions, letter, social media, email campaigns, etc.) to exert public pressure on authorities (e.g., journal editors, colleagues, university presidents, or provosts) that can sanction or punish the offender. Frequently enough, these organized campaigns manifest as petitions making some call to action (e.g., cancellation of an event; calls for termination; disinvitation of an invited speaker; and retraction of a published peer-reviewed article) that are then often signed by hundreds, sometimes even thousands of academics. Such groups may only be loosely organized and often feed off of one another's outrage, denunciations, and moral grandstanding on social media platforms, such as Facebook or Twitter [35]. On rare occasions an outrage mob may resort to more drastic tactics such as shouting down a speaker at an event, verbally threatening the offender, or getting physically aggressive with them (see [36] for a summary of such events).

For simplicity, in this paper, we refer to all such activities as those of "outrage mobs," and we define an outrage mob as:

> "A group or crowd of people whose goal is to sanction or punish the individual, individuals, or organization they consider responsible for something that offends, insults, or affronts their beliefs, values, or feelings. This group or crowd demonstrates a flagrant disinterest in any further explanation from the target or targets and attempts to carry out punishment often by enlisting authorities with the power to level sanctions on the target or targets".

It is important for us to make some additional distinctions about outrage mobs and their relationship to scholarship suppression. An external outrage mob is made up of individuals outside of the academy who are not engaged in formal scholarship that targets a scholar (i.e., academic) or group of scholars (i.e., academics) for scholarship suppression. An external outrage mob cannot level sanctions or punishments on the offending person or group of people; however, we suspect that one of the goals of most, if not all, external outrage mobs is to convince those who do have such authority to sanction or punish the offending scholar. An internal (or academic) outrage mob functions in an almost identical way to an external outrage mob, except that it is made up solely of scholars who are targeting another scholar or a group of scholars for scholarship suppression. Finally, sometimes an external outrage mob succeeds in persuading some academics to lend support to their campaign. We refer to these instances, when they occur, as a hybrid outrage mob. Thus, free speech can come into conflict with free expression and academic freedom because people can make use of their own freedom of expression to organize outrage mobs, create petitions of denunciation, and the like, to suppress other people's free expression.

### 4.3. Witch Hunts and the Politics of Heresy

Our concept of an outrage mob draws heavily on the sociological scholarship on political witch hunts [13,14], the politics of heresy [15], and moral panics [13,14]. This scholarship explores the

identification of deviance, or heresy, and how this identification helps a community define the boundaries on what beliefs they consider acceptable and true. Bergesen [37] synthesized elements from the sociology of religion (e.g., [38,39]) and the sociology of deviance (e.g., [37]) to suggest that witch hunts represent a way for a community to establish where its moral boundaries are and to reaffirm what that community holds sacred. To do this, witch hunts manufacture deviance within a community by transforming a trivial activity (e.g., publishing a peer-reviewed article on gender differences in career preferences) into an action that is imbued with larger historical forces (e.g., reinforcing the oppressive patriarchy). This process can involve different levels of community mobilization, from simple verbal accusations and charges to the empowerment of members of the community as righteous agents on guard for evidence of subversion and deviance [13]. Suspected deviants are then accused, subjected to some sort of "trial" in which the outcome is fore-ordained and typically made to confess their "crime" or "crimes" (see [13,15]; see also [18]).

In a similar vein, Kurtz [15] argued that heresy plays an essential role in allowing people to clearly and systematically articulate what is considered acceptable and possibly (or even likely) true in a belief system, and what is considered unacceptable and therefore false and dangerous. Heretics are deviant members of the community, and the social construction of a heresy typically occurs during a status conflict within a community. Through the identification of heresy and the pursuit of heretics, a community defines or reaffirms its moral boundaries about what is permitted and what is not. This process can also give members of the community the ability to enhance their social status while also maintaining and reinforcing the community's existing power structure ([15]; see also [14]).

### 4.4. Moral Panics and the Construction of Deviance

Our concept of an outrage mob also draws from the sociological literature on moral panics (see e.g., [25,40]), which itself has some similarities to the scholarship on witch hunts. Goode and Ben-Yehuda ([25], p. 29) described the concept of moral panics as "likely to clarify [the] 'normative contours' and 'moral boundaries' of the society in which they take place". A key component of a moral panic is the creation of a folk devil or suitable enemy who is considered responsible for some threatening or damaging behavior or condition (see [40]). Cohen [40] argued that a folk devil has been stripped of any favorable characteristics and that the negative characteristics that remain often develop into exaggerated unfavorable stereotypes. In other words, a folk devil is a member of the community who has been identified as a deviant and a threat by a notable portion of a community or society.

The criteria for considering an event a moral panic are quite clear (for a thorough review of all the criteria, see [25], pp. 37–46). Fairly suddenly, some portion of a community or society agrees that some behavior of a certain group or category is concerning, problematic, and even harmful to others. This consensus among members of the community, that some other portion of the community is deviant, permits them to treat the deviants, or folk devils, in their midst with hostility and punish them. Yet, the concern about the threat posed by the deviant members of the community is always more substantial than a realistic appraisal of the situation would conclude. Thus, we are not arguing that every instance of a scholar, or idea, being targeted for suppression by an outrage mob qualifies as an example of a moral panic. Instead, what we are arguing is that outrage mobs that target scholars for their scholarship often demonstrate one or more of the elements of a moral panic.

For instance, consider the case of Selina Todd, a Professor of History at Oxford University's St. Hilda's College, and a feminist. In 2019, Todd was invited to speak at the Oxford International Women's Festival held at Exeter College. She was the target of protests by transgender activist groups and several speakers withdrew in protest. The organizers then deplatformed her. Later, in 2020, she was invited to give a keynote address at the University of Kent, which sparked another outrage mob, this time, primarily academics, denouncing her. The large number of signatories on the letter represents some level of consensus about Todd's views among members of that portion of the academic community. The talk was eventually postponed due to the coronavirus outbreak, and it remains to be

seen whether she will be allowed to give it [41]. Along the way, she was the target of threats of physical violence sufficiently credible and severe that her college provided her with security guards [42].

The signatories of the denunciation letter directly referenced one page on Todd's website (https://selinatodd.com/my-feminism) where Todd stated that she is opposed to amending the law in the United Kingdom so that people would "be able to define themselves as men or as women simply by describing themselves as such". Todd gave 3 reasons for her views:

1. There is a need for women-only spaces because of past violence against women by natural born males;
2. The need to collect robust data on sex-based participation in a variety of professional domains so that discrimination on the basis of sex can be identified and hopefully remedied and;
3. Since the notion that someone could "feel feminine" may reinforce conservative gender stereotypes about femininity by overlooking that what is defined as "feminine" or "masculine" has changed over time.

Todd concluded by noting her membership in and support for Woman's Place UK (WPUK), a woman's rights group concerned about how the push to replace biological sex with gender identity may obfuscate current levels of sexism against women in a variety of domains.

The denunciation letter contends that Todd's views, and those of WPUK, about biological sex are tantamount to questioning the right of transgender women to self-identify. We are not adjudicating between these two different positions in the present paper, though we believe research into the veracity of both positions is a more appropriate way to resolve such a matter compared to a letter of denunciation. We do, however, think the case of Selina Todd helps demonstrate how academic outrage mobs sometimes share some similarity with moral panics.

It is clear that there was concern about Todd because of her views—they were described as not only "problematic" and "unpopular", but "harmful". These sorts of criticisms do not address whether she was actually factually or empirically wrong in any of her claims; and the terms "unpopular" and "harmful" are actually empirical statements that, at least hypothetically, could be tested, could be found to be either correct or incorrect, and do not need to be assumed. In the absence of such evidence, we do not know that the criticisms were actually wrong; instead, they can be described as evidencing a flagrant disregard for facts or evidence bearing on her statements. Furthermore, and perhaps even more important, "harm" is tantamount to calling her ideas "dangerous", and dangerous ideas are explicitly protected under the principle of academic freedom.

It was further argued that giving her a platform would force trans and nonbinary members of the university community in the position of having to defend their right to exist. This was presented as justification for the denunciation. It is manifestly false because trans and nonbinary members could simply have ignored anything Todd presented.

The final, and most important, criterion for declaring something a moral panic is disproportionality. To meet this criterion, the threat or the danger posed by the scholarship must be exaggerated beyond what a realistic appraisal could sustain. In the specific case of Selina Todd, a perusal of her deplatformed but planned remarks for the Oxford International Women's Festival [43] demonstrates they were rather brief. She intended to speak fondly of how her parents met at Ruskin College and how because of the Women's Liberation Movement, and that first inaugural conference at Ruskin College 50 years ago, she grew up in a much different world than her mother did. She concluded by arguing that, despite the victories won by feminism over the past 50 years, there is still much work to be done for women's rights. We leave it to readers to decide for themselves if the deplatforming of Selina Todd was a proportional or disproportional response.

### 4.5. Scholarship Suppression vs. Other Reasons for Punishment

There are of course bona fide reasons to punish academics, such as criminal behavior, incompetence, harassment of students or colleagues, and scientific fraud and misconduct. Since such reasons are for

unethical or criminal behavior, rather than promotion or publishing of scholarship, they are beyond the scope of the present paper.

A grey area involves ideas expressed through means other than scholarship. A rising influence on firings or pressures to resign comes from targets' social media presence, in places such as Twitter or Facebook. Faculty are routinely targeted by outrage mobs for offensive tweets or posts on other platforms (see "white genocide" discussed previously). Another was fired for a slew of racist comments, including referring to President Obama as a "monkey" [44].

In this paper, therefore, we put aside these grey areas, because even though conventional academic ideas may play some role in them, they are also some witches' brew of snark, hatred, sarcasm, insult, and racism. Our focus was on suppression of ideas as expressed in scholarship. By scholarship, we included peer reviewed articles, books, book chapters, and blogs. We also included teaching, talks, colloquia, panel presentations, essays, editorials, and op-ed pieces that focus mainly on presenting the scholarship, and its implications, for a lay audience. We excluded social media posts (Twitter, Facebook, and Instagram) and message boards.

Furthermore, if one goes back far enough in history, one can find ample evidence of academics being punished, including being fired, for their leftwing views, a phenomenon that probably reached its zenith during McCarthyism [45]. However, our review focused on the current state of academia and was not meant to provide a broadly historical review. Therefore, we focused exclusively on punishment occurring in the last five years.

## 5. Emerging Trends in Scholarship Suppression

There are two main types of scholarship suppression: suppression by others and self-suppression. The most direct route to scholarship suppression in academia is to attempt to punish people for their ideas (suppression by others). The most obvious manifestations of the modern toolbox of idea punishment include: firing, loss of position (e.g., a dean is removed though may remain on the faculty), deplatforming, and retraction of published papers for anything other than fraud, misconduct, or flagrant and frequent data errors. If successful, the ideas being promulgated will be suppressed. A retracted paper is no longer in the literature; a deplatformed speaker has lost a platform, and a fired scholar may never return to academia or publishing.

However, even if punishment is "unsuccessful" in the sense that none of those actually occur (the target is not fired, deplatformed, and their paper is not retracted), it may nonetheless be highly effective at suppressing ideas for several reasons:

1.  Defending one's self from such attacks is potentially time-consuming, emotionally exhausting, and, in some cases, may be quite expensive if lawyers get involved [46]
2.  The time and effort spent defending one's self from such attacks is time not spent engaging in scholarly activities; therefore, the productivity and ability to influence discourses and canons in the field in which the target works are reduced.
3.  The targeted scholar, even after successfully fending off the attack, may decide that whatever constituted the basis for the attack, and anything like it, is just not worth the grief that comes with pursuing it.
4.  Others, especially younger scholars seeking jobs or tenure witnessing the event may reach a conclusion along the lines of "the guild of professionals to which I aspire to join has declared certain types of work worthy of sanction, so maybe I should just work on something else".
5.  The attack may successfully sully the target's reputation, even if the target is not otherwise punished. Given that academia is a social reputational system, this can be quite enough to create formidable obstacles to getting ideas platformed, published, or funded.

### 5.1. Self-Suppression

Self-suppression occurs when people do not pursue certain ideas or try to publish certain findings because they fear punishment or prefer that the findings do not see the light of day. Self-suppression is notoriously difficult to empirically assess because there is mostly an absence of evidence (if the idea is suppressed, it cannot usually be found). This dynamic was captured beautifully in a podcast by social psychologists Michael Inzlicht and Yoel Inbar [47]. Inzlicht, at about 25 min in:

> "What if I felt that overemphasis on oppression is a terrible idea, hurts alleged victims of oppression, and is bad for everyone? What if I was outspoken about this? I suspect I would face a lot more opposition. Even though not much could happen to my job security, I'd have a lot of people screaming at me, making my life uncomfortable. And, truly, I wouldn't do it, because I'd be scared. I wouldn't do it because I'm a coward".

Our view is that Inzlicht is less of a coward than most simply by virtue of having gone forward with this podcast including this sort of statement. Nonetheless, it also nicely captures the social psychology of scientific idea self-suppression.

However, we know of at least 17 cases of self-suppression. Zigerell [48] discovered 17 unpublished experiments on racial bias embedded in nationally representative surveys totaling over 13,000 respondents. These unpublished experiments failed to detect evidence of anti-black bias among white respondents but did detect pro-black bias among black respondents.

Another example of self-suppression can be found in IAT (Implicit Association Test) research. In response to criticism of the ability of IAT studies to account for racial discrimination [49], a retort emphasized the validity of the IAT and included in its title: " ... Executive Summary of Ten Studies that no Manager Should Ignore" [50]. Putting aside the fact that six of the ten studies did not address racial discrimination, even the four that did, found almost no evidence of racial discrimination (see [51], for a review). This was simply not reported in Jost et al.'s reply [50], or in any paper we know of that has cited that reply, until we did a deep dive into the 10 studies and discovered the almost complete absence of racial bias effects [52]. Of course, it is possible that, rather than suppression, no one considered it relevant. However, how could findings showing little or no bias not be relevant to establishing the importance of the IAT to predict racial bias?

This raises the following unanswerable questions: How many other unpublished studies providing quality data relevant to important social issues and controversies are out there that have gone unpublished because the researchers feared repercussions, did not see the value of reporting it, or themselves did not want the results to become widely known? How many published papers have buried results (either their own or, in the case of reviews, others) in order to avoid highlighting findings that conflict with preferred narratives? The answer is currently unknowable, but it almost surely exceeds that described.

### 5.2. Suppression by Others: Modern Examples of Academics Targeted for Punishment for Their Scholarship

What type of scholarship evokes punishment by academics? Most of these cases involved findings or arguments that challenged (or, perhaps, threatened) academics' equalitarian sensibilities (race, sex, ethnicity, colonialism, etc., [1]). We were interested in identifying examples of academics advocating ideas in their scholarly work (as just described above) that offend rightwing sensibilities and who were subject to punishment by academic outrage mobs but were unable to do so. To be clear, there are ample examples of rightwing outrage mobs targeting leftwing academics for their political statements, especially on social media (e.g., [2]), but we excluded those because they were mobs organized outside of academia, and were often for social media posts rather than for scholarship. Additionally, as discussed previously, hundreds of academics lost jobs during the Red Scare of the 1940s and 1950s. Nonetheless, we were unable to find a single example over the last five years of an academic sanctioned by other academics for violating rightwing sensibilities regarding their scholarship.

5.2.1. Firings, Non-Renewals, and Forced Resignations

Firings since 2016 include (see [1] for more details):

- Alessandro Strumia, physicist working for CERN, fired (technically, not renewed, in 2018–2019), after presenting a data-based talk arguing that women were not discriminated against in physics. Although multiple issues may have contributed to his non-renewal, he was denounced primarily for his ideas.
- Noah Carl, social scientist, had accepted a postdoctoral position at St. Edmunds College (United Kingdom), which was ultimately rescinded in response to a petition denouncing him on these grounds: "A careful consideration of Carl's published work and public stance on various issues, particularly on the claimed relationship between 'race', 'criminality' and 'genetic intelligence', leads us to conclude that his work is ethically suspect and methodologically flawed". We note that the petition did not actually identify any methodological flaws and that the commission of inquiry tasked with evaluating his scholarship reached this conclusion: "Dr. Carl was ... an extremely strong candidate indeed having performed with conspicuous success at every academic stage ... [and] was the unanimous choice. No-one else impressed to anything like the same degree".
- Allan Josephson, Professor of Psychiatry at University of Louisville was demoted then fired after being denounced for making this comment at a conference: "When treating children with gender dysphoria, medical professionals should first seek to understand and treat the psychological issues that often cause this confusion before pursuing more radical, aggressive treatment".
- Susan Crockford, University of Victoria. She had an unpaid affiliation with the University for 15 years, which was not renewed in 2019, after she published a book arguing that, contrary to popular environmental narratives at the time, there was no ongoing devastation of polar bears, leading to her denunciation as a "climate denialist". It is interesting to note that polar bear population estimates have gone from 20,000–25,000 in 2012 [52] to 22,000–31,000 in 2019 [53].

5.2.2. Punishment Other than Termination

Academic outrage mobs can inflict all sorts of punishment that falls well short of firing, but which are certainly noxious experiences for their targets (see our prior discussion of Selena Todd). More details about these examples, including links to additional blogs, petitions, and new stories, can be found in Jussim (2020) [1].

- Bruce Gilley, Political Science, Portland State University, 2017. His paper, The Case for Colonialism, was retracted after academics initiated a petition calling to retract, signed by thousands, and then both Gilley and the journal editor received what they considered to be credible death threats.
- The National Association of Scholars, perhaps the last right-leaning academic organization in all of U.S. academia, held a conference in 2020 titled "Fixing Science". It was denounced as a shill for conservative and corporate interests promoting climate change denialism. There were also social media and email campaigns that pressured invited speakers not to attend. Although most did attend, two early career scholars withdrew. Whether this was because they earnestly believed in the validity of the denunciations, or were simply intimidated and feared for their careers, or some mix of both, remains unclear.
- Stephen Gliske, a neuroscientist at University of Michigan, published a paper presenting a new theory of the development of gender dysphoria. It offended trans activists and their academic allies, who launched a retraction petition that was ultimately successful.
- Ted Hill, Math professor, Georgia Tech, wrote a paper offering an evolutionary explanation for the male variability hypothesis (the idea that human males are more variable than human females on many attributes). It was accepted for publication at a journal; this evoked protests and outrage, which had the effect of pressuring the accepting journal to "unaccept" the article. He then had it accepted at another journal, which evoked more outrage (the manifest substance of

which involved the process by which the paper was accepted), and it was again unaccepted [54]. The paper remains unpublished as far as we know.

- Kathleen Lowry, a feminist professor of anthropology, lost her position as the undergraduate associate chair in 2020 for claiming that biological sex exists and is important, on the grounds that "it was not in the best interests of the students" for her to continue in the position.

There are many more examples like these that can be found in Jussim (2020) [1]. Furthermore, the pace at which this sort of punishment is occurring seems to have increased over the last five years compared to any time prior, and especially over the last year compared to even the prior few years.

5.2.3. The Importance of Authorities in Resisting Outrage Mob Calls for Punishment

Although outrage mobs often trigger the punishment process, in Western democracies, mobs no longer actually burn witches at stakes. For most punishment to occur in academia, some authority has to agree to implement the mob's punishment. That is, mobs do not get academics fired; it is high level administrators, such as deans, provosts, and university presidents that implement firings. Mobs do not get papers retracted; that is the decision of editors and editorial boards. Thus, the key turning point in whether an academic outrage mob is effective at punishing an academic for their ideas is usually the action of authorities. Although as we articulated earlier, simply being targeted by an outrage mob will likely be experienced as punishment, we also acknowledge that there is a qualitative difference to being denounced publicly and extensively (however unpleasant that may be) and being actually fired. It is the authorities, not the mob, that decide whether to inflict punishment that goes beyond social approbation.

Lest we paint too bleak a picture of academia as an institution overrun with dogmatic ideologues and spineless quislings unwilling or unable to defend academic freedom and free inquiry, we described several cases here of authorities not caving in to mob pressure. Again, summary descriptions of these events, including links to more detailed stories and denunciation petitions and letters, can be found in Jussim (2020) [1]:

- An academic outrage mob petitioned (July 2020) to have Professor Steven Pinker, Psychology, Harvard, removed from the Linguistic Society of America's list of distinguished academic fellows and their list of media experts. There were a variety of vague allegations. This petition failed. It was so obviously filled with falsehoods and misrepresentations that numerous sources were able to debunk its charges.
- Philosopher Rebecca Tuvel [55] published a paper in *Hypatia*, a leading feminist philosophy journal, titled "In Defense of Transracialism", in which she argued that people could choose to identify as whatever race they preferred. She drew on common postmodernist ideas suggesting that race is not an essentialist or biologically determined category and that it is socially constructed. Just as people can, according to this view, identify as any gender, she argued that the same perspectives would mean they could also do so for race. The paper was denounced by hundreds of academics who signed an open letter calling for retraction, including the claim that Tuvel caused "harm and violence". *Hypatia*'s board of directors stood firm and refused to retract the article.
- Littman [56] published preliminary evidence for "rapid onset gender dysphoria", which referred to the idea that, for some adolescents, identifying as a different gender seemed to have been something that emerged suddenly, more or less "out of the blue", rather than from a longstanding history of identifying differently than the sex one was assigned at birth. The paper was quickly denounced by transgender activists claiming the paper caused "harm" and "denied their identities". From here, the story took some strange turns. The journal publishing the paper (PLoS One): 1. instituted post-publication review; 2. apologized for their handling of it; 3 persuaded Dr. Littman to revise the paper; and 4. published the revision as a "correction". The "correction" was particularly odd because there were no errors identified in the original, and no factual changes. Instead, Littman added some context that qualified her claims and conclusions. Although Dr. Littman was fired

from an ancillary consulting position, this paper was not retracted. Thus it is included here as an example of an authority (in this case, the editors) resisting an outrage mob's call to retract, although the incident is plausibly considered an intermediate case, because she was made to jump through extraordinary hoops that, as far as we know, no other author has ever had to jump through at PLoS journals.

- Dr Abigail Thompson [57] published an editorial criticizing the use of mandatory diversity statements in academic hiring. This triggered an academic outrage mob denouncing her and a petition calling for removal from her position as vice president of the American Mathematics Society (AMS). According to the AMS website (https://www.ams.org/about-us/governance/officers/officers), as of September 2020, she was still listed as vice president. Thus, although we had no inside information, the AMS did not cave to mob outrage. An interesting epilogue is that she has also received a Hero of Intellectual Freedom Award from the American Council of Trustees and Alumni [58].

## 6. Conclusions

This paper reviewed theoretical perspectives on and emerging trends in the suppression of academic scholarship. It provided several unique contributions to understanding suppression. First, we reviewed legal, moral, and social issues in distinguishing among free speech, free inquiry, and academic freedom. Second, we showed that, because academia is a social reputational system, it is especially vulnerable to groupthink and a stifling intellectual conformity, despite lip service or even bona fide protections resulting from academic freedom statements and tenure.

Third, we pointed out that protections and freedoms such as free inquiry and free expression, which are often used in ways plausibly viewed as interchangeable in much social discourse, can and are often in tension and conflict. The premier case is that of a denunciation mob; they are exercising their legal free speech rights, and, if they are successful, they will suppress certain ideas. We then reviewed theoretical perspectives on and the history of moral panics, which have long played a role in suppressing certain ideas. Finally, we reviewed a series of real-world cases where academics have been fired, successfully punished in ways short of firing, and targeted unsuccessfully for punishment by outrage mobs for their ideas.

### 6.1. Limitations

Nonetheless, there are certain important limitations to the perspective presented here. Although academic censorship may be on the rise, in historical and international terms, this still pales in comparison to what has often occurred under religious, monarchic, or authoritarian regimes. Current censoriousness, at least in the U.S., is also less severe than it was during the height of McCarthyism and the Second Red Scare (roughly 1947–1960), when hundreds of faculty lost their jobs [11]. Although that era may have seemed like government censorship, in fact, it was the universities themselves that did most of the firing. Put differently, administrators and many faculty enthusiastically embraced purging communists from their ranks [10].

Nonetheless, if academic censorship is on the rise, it raises the question, "Why now?" Our review was only intended to describe the social and psychological processes by which such censorship occurs. Therefore, an important limitation of the present perspective is that it has not sought to explain why it seems to be increasing now.

In speculation, we suggest that several forces may have come together to contribute to this rise in calls for censorship and suppression, at least in the U.S. and possibly elsewhere. First, many analyses indicate a recent dramatic rise in what is sometimes called tribalism [59] or affective polarization [60]). These concepts refer to an intense identification with and emotional attachment to one's political ingroups, along with a concomitant tendency to despise, derogate, and demonize one's opponents [61–63].

Along with tribalism and polarization, some have argued that there has been an increasing sacralization of victimhood (e.g., [64]). Combine these developments with a number of disciplines in academia skewing left in their politics [28,65,66], then the demonization of academics who are perceived to threaten groups viewed by many of those on the left as warranting special protections (marginalized groups, stigmatized groups, etc.) may seem almost inevitable. Of course, demonization by itself is not quite enough to produce censorship. The final ingredient, which may act more like pouring an accelerant on an already smoldering fire, may be social media. Platforms like Facebook and Twitter have enabled outraged academics to quickly organize into large mobs through open letters, email campaigns, electronic petitions calling for sanctions, and public denunciations. Authorities (editors and administrators) may either be persuaded or concerned about their organization's public image, and decide it is either justified or simply easier to sanction the target than to have to deal with a mob or "explaining" to the public why they are defending a person others have publicly denounced as despicable.

### 6.2. What Should Be Done?

A most pressing question, unaddressed in this paper, is what should be done about this state of affairs? We concluded our paper not with answers, but with an attempt to begin a conversation about how to address this emergent social problem.

One possibility is nothing should be done, which would be the recommendation if one believes this is an excellent state of affairs. This is the position one would adopt if one believes some academics deserve the authority to prevent others even from airing ideas, such as the existence of rapid onset gender dysphoria, the reality of biological sex, or that polar bear populations are actually rising.

On the other hand, some may believe this is a bad state of affairs, and that no one should have the authority to prevent others from getting a hearing for their ideas. In that case, what can be done to prevent scholarship suppression? Our review suggests that some answers might involve creating disincentives to suppress by appeal to authorities. What might such disincentives be? Some possibilities:

1.  The most obvious is financial: can these mobs be sued for defamation? We doubt it, at least most of the time. The U.S. Courts, for example, have repeatedly decided that publicly denouncing someone as racist is simply opinion, and, therefore, fully protected speech [67]. On the other hand, at least one defamation suit in Canada was successful in evoking payment of expenses and a public apology after unjustifiably referring to a reporter as a "neo-Nazi" [68]. If enough such suits were successful, the threat of such a suit might become a more effective deterrent. However, as discussed, the mere possibility of having to hire lawyers to fight defamation suits may be sufficient to deter some outrage mobs from even getting started.

2.  Another possibility is to exploit the academic social reputational system itself—what if the tactics of the outrage mob were turned on the leaders and organizers of such mobs? What if their reputations were impugned and their employers targeted with emails and petitions denouncing them in ways not readily refutable (say, as authoritarian bullies?). If enough "counterattack mobs" succeeded, again, the mere potential to have one's career damaged by engaging in attempts to suppress others' work may be sufficient to deter some such attempts.

3.  Yet a third possibility is to, somehow, pressure the relevant authorities to actually uphold their responsibilities to protect academic freedom. Most colleges and universities at least pay lip service to academic freedom, and many have written documents testifying to such commitments. In the case of academics, the key authorities, then, are usually the administrators (chairs, deans, provosts, presidents, etc.) who have ultimate responsibility for deciding whether or not to sanction faculty accused of some sort of heresy by an outrage mob.

Although these are three hypothetical options, none seem particularly likely to actually solve the problem of suppression by academic outrage mob. Suing is quite difficult and, in the U.S., not likely to succeed. Attempts to denounce the denouncers are a recipe for the escalation, not the dialing back,

of outrage mobs. Additionally, given how regularly authorities have caved to these mobs in the past, we see no simple path to changing their behavior, as desirable as that may be.

Academia is littered with cases of influential papers and ideas having initial difficulty getting published. However, the work is not suppressed if the scholar can revise the paper, make it stronger, and attempt to publish it at the same or different outlet. Scholarship suppression has a superficially similar appearance to rejection—in both cases, scholarship does not see the light of day. However, there is a key difference, when scholarship is suppressed, it does not get a fair hearing. It is denounced and rejected, never submitted, or never conducted in the first place. Rejection is quality control, however imperfect; suppression impoverishes the knowledge base. Academics, administrators, and the broader public should think long and hard about whether scholarship suppression is a fundamentally desirable or undesirable feature of the quest for knowledge.

**Author Contributions:** Conceptualization, S.T.S. and L.J.; writing—original draft preparation, S.T.S., L.J., and N.H.; writing—review and editing, L.J., N.H., and S.T.S.; project administration, L.J. All authors have read and agreed to the published version of the manuscript.

**Funding:** This research obtained no external funding.

**Conflicts of Interest:** The authors declare no conflict of interest.

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
