# Peer review of "Scholarship Suppression: Theoretical Perspectives and Emerging Trends"

_societies, doi:10.3390/soc10040082_

Round 1
Reviewer 1 Report
The subject matter of this paper - scholarship suppression and the emergence and role of denunciation mobs in this - is very important and I am very pleased to see the efforts being made to analyse this. The authors have a lot of interesting thoughts and have collected plenty of material and examples. However, there are problems which mean revision is needed.
One is that the detail of the main example used in the section on witch hunts and heresy is not correct. This is the discussion of Selina Todd, and what happened to her in 2019 / 2020 where she was invited to speak at different events. The text on p10 about the University of Kent and Ruskin College is very mixed up. This is not entirely the author's fault, because this paragraph in the article in Medium they has used as a source is wrong.
'Professor Selina Todd, history professor at Oxford University, was disinvited/deplatformed from a feminist conference at Oxford’s Exeter College on February 29, 2020. This resulted from protests by trans activists, which included this open letter from hundreds of academics calling for her to be dropped from the program. The talk she would have given, had she been permitted to give it, can be found here.'
What actually happened in Oxford in early March 2020 is discussed here: https://www.bbc.co.uk/news/uk-england-oxfordshire-51737206
The story with the planned lecture at the University of Kent is different. This was an invitation from the School of English to give the final lecture in a series. There were objections and the University was pressured to rescind the invitation (that is the letter that the paper discusses). However, the University did not concede to the demands made in the letter, citing the University's commitment to free speech. Further, public support was offered by academics at the University, in the School of English and in many other Schools, in support of Todd coming to do the lecture. See comment here: https://www.afaf.org.uk/will-academic-freedom-be-upheld-at-the-university-of-kent/
The lecture at Kent University would have gone ahead; it was never cancelled. However it was planned for March 18 and the whole University closed that week due to the pandemic.
This does not necessarily take from the drive of the argument, around some of the issues and the critique of the case made for no platforming on the basis of 'harm'. But these arguments were not accepted by all and the outcome was not as the article suggests.
The larger issue is the place of the analysis in this section, and then afterwards, in the article as a whole. It seems as though this is supposed to the be the main contribution of the piece i.e. to explain why suppression of views is happening. However, the analysis remains rather descriptive. There is a lot of detail and example, but there is not enough depth sociologically, of the case for the article is its contribution in explaining these trends to suppression from a sociological perspective.
Anchoring this in the literature on witch hunts and moral panics might be successful but the explanation would need to go further. The deficit presently is that there is no account provided of why this is happening now i.e. if suppression is to be thought of as a species of witch hunt and moral panic, then how has morality shifted to make it possible? Why have these events developed and expanded in the way they have? The authors usefully refer to the matter of boundaries and how and why they are drawn, and this thought could perhaps be taken a lot further, in regards to how a new sort of boundary is emerging in a context where the general case if for dismantling borders and boundaries. The recent book by Furedi on the subject is very useful on this (https://www.waterstones.com/book/why-borders-matter/frank-furedi//9780367416829)
Overall, I think the article needs to be revised to make it less bitty. After the section on moral panics and witch hunts, which seems to be the main focus for the sociological contribution, the text becomes rather note-like, with lists and bullet points around examples. It takes more than 8 pages to get to this part. The text in these opening pages is interesting and raises a lot of thoughts. However, it's made up of a lot of short sections and there is not enough of an overall case or argument that takes the reader somewhere i.e. sets the context for the points about the sociological analysis of suppression (if that is what the meat of the article is intended to be). One specific point is also the synthesis of the discussion in these opening parts (which is focused explicitly on the US, and addresses via J.S.Mill, a range of ideas and their relation to legal precepts in the US)and the fact one key example later is about Selina Todd events in England.
In sum, I think the issues discussed in the article are very important, the authors have done a lot of good work and have some very good idea and examples, but the article does not presently flow well and hold together. it needs a central question or focus around which it its built - unless it is presented differently, as some 'notes on...'.
Author Response
We sincerely thank Reviewer 1 for identifying our flawed description of the Selina Todd episodes. Our original description erroneously treated two separate incidents as a single case. We went back to the original news reports, including the ones linked by the reviewer, and thoroughly rewrote this section. Upon review of the actual events, it was even better (as an example) than we had originally captured. It was two, back-to-back, denunciations and attempts to deplatform. Therefore, we expanded this section to include both denunciations.
Having caught that error, we wondered whether there were others. So we went back and double-checked each and every one of the other examples we included in the paper. All checked out fine and prompted no major revisions.
About the early sections of the paper, the reviewer believed there was not a clear thematic point to them. We believed they were mission-critical. In our experience, outside of the scholars focusing on these issues, many people seem to confuse the legal versus moral issues surrounding free speech and inquiry, and academic freedom and free inquiry are often not clearly distinguised from free speech per se. Therefore, we believed it was essential to draw these distinctions before proceeding to the social and psychological processes of suppression -- which rarely involve legal infringements on free speech.
We agreed with the reviewer that we had not provided insight into "why now?" We have added a section to the Conclusion on Limitations that acknowledges this and other limitaitons. Unfortunately, both the short timeframe for revision and the fact that paper is already over 35 pages of text mitigated against adding a full section attempting to address this question. Nonetheless, we have added some speculations in the Conclusion regard "why now?"
We are sincerely grateful to the Reviewer for comments that have substantially improved the paper.
Reviewer 2 Report
Dear Authors,
The manuscript entitled “Scholarship Suppression: Theoretical Perspectives and Emerging Trends” deals with an interesting, important and current topic that has to be widely discussed in the scientific community. However, the authors should consider the following comments to improve it.
First of all, the structure of the manuscript is not always clear; although there is section 1, numbering of the subsequent sections is missing, making it difficult to trace the hierarchy of sections. Related to this, there are heading-like sentences on p. 2 and p. 13, which might not supposed to be headings. Moreover, the first section should be titled “Introduction”. To make the text easier to follow, it is recommended to indicate in bridging sentences what will follow and why.
There are several typos in the text, e.g.:
- “0.” on p. 1.
- Inconsistent use of capital letters in headings.
- Unjustified use of the full stop at the end of headings.
- Unnecessary use of spaces throughout the text.
- Numbering of Todd’s reasons on p. 10 is wrong.
- Stevens, Jussim, Anglin, Contrada et al., 2018 is missing from the references list.
- IAT should be mentioned fully in brackets (p. 12).
- The paragraph about Alessandro Strumia is an example, thus a bullet point is missing here (p. 13).
- In contrast, the paragraph starting in line 677 is not an example, thus there is no need to use a bullet point here.
- Author Contributions, Funding, and Conflicts of Interest are not indicated.
- Bergesen appears twice in the text with year 1978.
- Flore and Wicherts (2015) is missing from the text.
Author Response
Reviewer 2 described the paper as "interesting and important" and declared that it "has to be widely discussed in the scientific community." We are, of course, grateful for these positive comments, and we sincerely hope that the reviewer is correct and this paper inspires that discussion.
In addition, Reviewer provided a slew of technical, typo, and formatting recommendations. We have corrected typos, added organizational numbering to the headers and subheaders and provided missing references (if they are still cited; some were removed entirely).
If there are any residual issues like these, of course we will correct them.
Reviewer 3 Report
This is a very useful paper with excellent detail about the issues regarding academic freedom. The legal focus is largely related to the United States whereas many examples are from the UK. It would be useful, when discussing the legal/political framework to give some indication of the state of play in the UK regarding these issues.
Author Response
Reviewer 3 was very positive about our paper, describing it as "a very useful paper with excellent detail about the issues regarding academic freedom." We certainly hope so!
Reviewer 3 also requested we go into more detail about the state of play in the U.K. We did this in a number of small ways throughout the paper. Our primary argument was that illegal infringements on academic speech by government entities has not been the primary source of scholarship suppression lately. Therefore, at least in this paper, it was important for us to distinguish legal infringements from more social or normative sources of suppression, primarily to be sure readers would clearly understand that we were not mostly focusing on legal infringements. Furthermore, although Sean Stevens, through his work with The Foundation for Individual Rights in Education, has some expertise with respect to legal protections for speech in the U.S., none of us had comparable expertise outside the U.S. Therefore, we lack the expertise to clearly and accurately describe the legal state of play in the U.K. However, we also viewed the moral issues surrounding free inquiry and academic freedom as fundamentally similar in the U.S., U.K. and other western democracies (we have at least two examples from Canada as well). We have made a large number of small revisions throughout the paper to capture this idea.
Round 2
Reviewer 1 Report
I am pleased the comments in the previous review were helpful in revision. It's a good paper and definitely opens up some important issues. I don't know if it's because the text was cut off because of word length, but the article is unfinished (it stops mid sentence at line 838). It is near the end I think, but that needs looking at. I accept the points made about what happens when reviewers ask for more and there is no space, but I did wonder about the Selina Todd case as also a positive example, because of how the University decided to respond: https://www.kent.ac.uk/news/uncategorized/24352/professor-selina-todd-lecture I think there are some difficulties with the US/UK combinations because of the differentail frameworks on all levels, and responses, but maybe these are questions for a further article.
Author Response
A few words were cut off from the very last sentence of the paper. That has now been corrected.
We concur with the reviewer that a comparison of the US & UK would be very valuable, but that it deserves its own paper (or at least a major section in a subsequent paper).